# Nitrate–Nitrite–Nitric Oxide Pathway: A Mechanism of Hypoxia and Anoxia Tolerance in Plants

**DOI:** 10.3390/ijms231911522

**Published:** 2022-09-29

**Authors:** Arbindra Timilsina, Wenxu Dong, Mirza Hasanuzzaman, Binbin Liu, Chunsheng Hu

**Affiliations:** 1Hebei Key Laboratory of Soil Ecology, Center for Agricultural Resources Research, Institute of Genetics and Developmental Biology, Chinese Academy of Sciences, Shijiazhuang 050021, China; 2Department of Agronomy, Faculty of Agriculture, Sher-e-Bangla Agricultural University, Dhaka 1207, Bangladesh; 3Xiong’an Institute of Innovation, Chinese Academy of Sciences, Xiong’an New Area, Baoding 071700, China

**Keywords:** denitrification, plants, hypoxia and anoxia, nitric oxide signaling, nitric oxide toxicity

## Abstract

Oxygen (O_2_) is the most crucial substrate for numerous biochemical processes in plants. Its deprivation is a critical factor that affects plant growth and may lead to death if it lasts for a long time. However, various biotic and abiotic factors cause O_2_ deprivation, leading to hypoxia and anoxia in plant tissues. To survive under hypoxia and/or anoxia, plants deploy various mechanisms such as fermentation paths, reactive oxygen species (ROS), reactive nitrogen species (RNS), antioxidant enzymes, aerenchyma, and adventitious root formation, while nitrate (NO_3_^−^), nitrite (NO_2_^−^), and nitric oxide (NO) have shown numerous beneficial roles through modulating these mechanisms. Therefore, in this review, we highlight the role of reductive pathways of NO formation which lessen the deleterious effects of oxidative damages and increase the adaptation capacity of plants during hypoxia and anoxia. Meanwhile, the overproduction of NO through reductive pathways during hypoxia and anoxia leads to cellular dysfunction and cell death. Thus, its scavenging or inhibition is equally important for plant survival. As plants are also reported to produce a potent greenhouse gas nitrous oxide (N_2_O) when supplied with NO_3_^−^ and NO_2_^−^, resembling bacterial denitrification, its role during hypoxia and anoxia tolerance is discussed here. We point out that NO reduction to N_2_O along with the phytoglobin-NO cycle could be the most important NO-scavenging mechanism that would reduce nitro-oxidative stress, thus enhancing plants’ survival during O_2_-limited conditions. Hence, understanding the molecular mechanisms involved in reducing NO toxicity would not only provide insight into its role in plant physiology, but also address the uncertainties seen in the global N_2_O budget.

## 1. Introduction

Oxygen (O_2_) deficiency hinders respiration and other biochemical processes essential for plants’ survival, but extreme events such as heavy precipitation and flooding cause waterlogging, which directly affects O_2_ supply and prevents their growth [1]. Moreover, several other conditions can also lead to hypoxic and anaerobic conditions in well-aerated tissues of plants. For example, pathogen attacks, tissue exposure to freezing, sulfur dioxide (SO_2_), ozone, and water deficiencies can cause anaerobic conditions, leading to anaerobic metabolisms in plant tissues [2,3]. Abiotic stress such as salt stress can disrupt the symplastic connections between cells, which decreases the permeability of cells to O_2_, resulting in hypoxia and anoxia [4,5]. Moreover, under normal conditions, an endogenously generated O_2_ gradient also exists, such that the O_2_ concentration may fall below 5% in plant tissues, such as in seeds, bulk tissues, shoot apical meristems, and roots [6].

Hypoxia and anoxia result in the modification of various normal metabolic paths [7]. Thus, they usually inhibit respiration, photosynthesis, nitrogen assimilation, biological nitrogen fixation, water and nutrient uptake, and stomata closure in plants [5,7,8,9,10,11] through a reduced adenosine triphosphate (ATP) concentration, nicotinamide adenine dinucleotide (NAD^+^) and nicotinamide adenine dinucleotide hydrogen (NADH) ratio (NAD^+^/NADH), and cell viability [12]. Meanwhile, the accumulation of reactive oxygen species (ROS) and reactive nitrogen species (RNS) is triggered, which severely damages the cell components [13]. Moreover, during hypoxia and anoxia, a drop in pH causes cytoplasmic acidosis which affects numerous metabolic activities that may even contribute to plant death [14]. Overall, hypoxia and anoxia have numerous deleterious effects on plant metabolism (Figure 1). The effects of O_2_ deficiency could be more severe to hypoxia- and anoxia-intolerant plants as compared to tolerant plants. For example, O_2_ stress directly reduces germination rate and coleoptile growth in barley, oat, and rice, while growth is more pronounced in anoxia-tolerant rice than in barley and oat [1,15].

To survive O_2_ deficiency, plants use numerous strategies through biochemical, anatomical, and morphological changes (Figure 1). However, the accumulation of ethanol and lactic acid (major products of the fermentation process) are toxic [16]. Moreover, antioxidant defense systems could also be a limiting factor if the stress is present for a longer time or beyond the tolerance capacity. This suggests that if anaerobic processes proceed for a longer time, the ultimate fate of plants is death. Along with metabolic changes, plant adaptation mechanisms can improve tissue O_2_ status. A number of mechanisms have been reported to help plants to improve O_2_ status during soil waterlogging conditions. For example, O_2_ distribution from aerial parts to roots is facilitated by the formation of aerenchyma [17,18], and its formation is faster in flood-tolerant than intolerant plants [18,19]. Similarly, adventitious root formation can also improve the O_2_ status of plants during waterlogging conditions [20]. Meanwhile, the balanced production of ROS and RNS and an increase in antioxidant activities can enhance tolerance to hypoxia and anoxia in plants [21].

Nitric oxide (NO), a widely recognized signaling molecule, plays an important role in hypoxia and anoxia tolerance in plants [22,23]. Not only NO but also nitrate (NO_3_^−^), nitrite (NO_2_^−^), and nitrate reductase (NR, EC 1.6.6.1) play a similar role in plants during O_2_ deficiency [24]. This suggests that tolerance to O_2_ deficiency is due to the reductive pathways of NO formation. However, numerous studies indicate that O_2_ deficiency, as well as other stresses, can trigger NO formation [23,24,25]. Meanwhile, a higher concentration of NO could be cytotoxic, leading to the accumulation of ROS and other RNS that would lead to nitro-oxidative stress [25,26]. Nitric oxide could promote [27] or inhibit [28] ethylene biosynthesis, a key phytohormone for plants’ survival during O_2_ limitation, while the latter case is mediated through the S-nitrosylation of methionine adenosyltransferase (MAT1) [28]. Phytohormones such as salicylic acid (SA), jasmonic acid, and abscisic acid (ABA) reduce oxidative stresses and enhance the activities of antioxidants during stress conditions [29,30]. However, NO is reported to inhibit the activities of antioxidants, as well as proteins involved in regulating phytohormones through S- nitrosylation [30,31,32], thus, again, increasing nitro-oxidative stress in plants.

Thus, there should be a fine regulation of these signaling molecules (NO, ROS, and other RNS) for beneficial roles. The key to surviving during hypoxia and anoxia depends upon mechanisms that could lessen the harmful effects of nitro-oxidative damages by increasing the activities of adaptation mechanisms. Therefore, understanding the reductive pathways of NO formation along with NO scavenging mechanisms would provide insight into the mechanisms involved in lessening the effects of O_2_ deprivation. Hence, the main aim of this review is to highlight the role of reductive pathways of NO formation, while emphasizing the NO scavenging mechanisms that could reduce the nitro-oxidative stress and increase the hypoxia and anoxia tolerance in plants.

## 2. Pathways of NO Formation during Hypoxia and Anoxia

Various pathways of NO formation in plant cells have been documented, and they have been categorized into oxidative and reductive pathways. Oxidative pathways are oxygen-dependent, involving L-arginine, polyamine, and hydroxylamine [33]. The reductive pathways of NO formation occur during low O_2_ and are dependent on NO_3_^−^, NR, NO_2_^−^, plasma membrane NR, plasma membrane-bound nitrite reductase (PM NiNOR), xanthine oxidoreductase in plant peroxisomes, photosynthetic-electron-transport-chain-dependent NO_2_^−^ reduction in chloroplasts, and mitochondrial electron transport chains (ETCs) such as cytochrome bc_1_ complex (complex III, EC 1.10.2.2), cytochrome c oxidase (CcO, EC 1.9.3.1), and alternative oxidase (AOX, EC 1.10.3.11) in mitochondria [33,34]. In *Chlamydomonas reinhardtii*, NR, together with nitric-oxide-forming nitrate reductase (NOFNiR), reduces NO_2_^−^ to NO [35]. Moreover, NO_2_^−^ can be reduced to NO in acidic pH without the involvement of any enzyme. NO production pathways during O_2_ deficiency and other stresses would be different. For example, salt stress can increase both oxidative pathways (l-arginine-dependent) [36] as well as the reductive pathways of NO production [37]. However, during O_2_ deficiency, NO is produced through the reductive pathways [33]. Interestingly, this occurs not only during O_2_ deficiency but many other biotic- and abiotic-stress-induced reductive pathways of NO formation. For example, salinity stress, water deficiency, UV radiation, freezing, pathogen attacks, and wounding can trigger NO production in plants [38,39,40,41,42,43], which could be due to the fact that these stresses could lead to hypoxia and anoxia in plant tissues, while its formation could be a defense strategy to survive harsh conditions.

Nitric oxide is formed in various cell compartments such as the cytosol, apoplasts, chloroplasts, peroxisomes, and mitochondria of plants through enzymatic or non-enzymatic pathways [33]. Nitric oxide production in various compartments of plant cells has numerous functions. For example, NO formed in chloroplasts can prevent the oxidation of chloroplastic lipids and proteins, while NO-mediated peroxynitrite (ONOO^−^) production may result in its damage [44]. Similarly, NO formed in mitochondria can protect its components, while NO-mediated ONOO^−^ production causes mitochondrial dysfunction [45]. As O_2_ limitation, as well as other stresses, leads to a higher level of NO formation through the reductive pathways, understanding the possible mechanisms for maintaining the optimum level of NO is also essential and is discussed in later sections.

## 3. Role of Nitrate and Nitrate Reductase (NR) during Hypoxia and Anoxia Tolerance

Nitrate is not only an important form of nitrogen (N) source to plants but also a signaling molecule [46]. It is usually a major form of N in aerobic soil, and its uptake by plant roots is achieved through NO_3_^−^ transporters [47]. After being uptaken by roots, NO_3_^−^ is reduced to NO_2_^−^ by an enzyme called NR in the cytosol or plasma membrane or stored in the vacuole or transported to shoots and leaves for subsequent reduction [16]. Under normoxia, NO_2_^−^ is transported to plastids/chloroplasts and is reduced to ammonium (NH_4_^+^) by nitrite reductase (NiR, EC 1.7.7.1). Then, glutamine synthetase/glutamate-oxoglutarate aminotransferase (GS, EC 6.3.1.2)/GOGAT, EC 1.4.1.13) assimilates NH_4_^+^ into amino acids. However, during hypoxia and anoxia, the NO_3_^−^ or NH_4_^+^ assimilation path to amino acid as well as NO_3_^−^ transport to the aerial parts is greatly reduced [48]. For example, O_2_ deficiency decreases NO_3_^−^ and NH_4_^+^ assimilation and N incorporation into amino acids in various plant species as compared to normoxia [49,50,51]. Although N incorporation into amino acids is inhibited during O_2_ deficiency, several pieces of research have shown that NR is highly activated and NO_3_^−^ is reduced to NO_2_^−^ [52]. Interestingly, NO_3_^−^ consumption by soybean plants in hydroponics systems was higher during hypoxia than normoxia [48], which suggests that more NO_3_^−^ is metabolized during O_2_-limited conditions in plant cells. Therefore, most of the derivatives of NO_3_^−^ might be lost to the environment in the form of gases from plants during O_2_ limitation.

Several previous studies have shown that NO_3_^−^ and NR are beneficial for hypoxia and anoxia tolerance. Germinating seeds generally experience hypoxic and anoxic conditions [53,54,55] due to the compaction and hindrance of O_2_ diffusion by the outermost layers of seeds [56]. Studies have reported that NO_3_^−^ is beneficial during seed germination. For example, supplementation or priming with NO_3_^−^ increases the viability of germination in seeds of various plants [57,58,59,60]. Light and temperature influence seed germination, while NO_3_^−^ can reduce the dependency on environmental factors such as light [58,61] and temperature [62] during germination. Moreover, NO_3_^−^ can promote germination in seeds during salt, metal, and heat stresses [63,64,65]. The mechanisms of seed germination by NO_3_^−^ might be due to NO production in cytosol and mitochondria through the reductive pathways [55]. Similarly, NO_3_^−^ has been shown to increase activities of antioxidant enzymes such as catalase (CAT, EC 1.11.1.6) and superoxide dismutase (SOD, EC 1.15.1.1) during the germination process [59], which could scavenge ROS, thus preventing oxidative damage and promoting germination. Seed germination is promoted during conditions with a lower level of ABA [66] and a higher level of gibberellic acid (GA) [67], while NO_3_^−^ supplementation leads to the upregulation of the ABA catabolic gene *CYP707A2* and GA biosynthesis gene *GA20ox1* [64], thus promoting seed germination by decreasing ABA and increasing GA levels. Although ethylene is widely reported to promote seed germination, the role of NO_3_^−^ on its biosynthesis is unclear. Endogenous NO_3_^−^ levels in germinating seeds drop significantly during the first 24 h post-imbibition, and the role of NO_3_^−^ and NR activity in anaerobic seed germination depends on NADH and NADPH [68], which could be due to the fact that NO_3_^−^ serves as an alternate electron acceptor [69], similar to bacterial denitrification [70]. This is supported by the fact that NO_3_^−^ and O_2_ limitation induces high levels of both NO and N_2_O emissions from plants [24].

Waterlogging reduces several nutrients in plants, affecting plant metabolism [71], while the supplementation of NO_3_^−^ increases the uptake of nutrients such as N, P, Fe, and Mn [72]. Nitrate can improve cytoplasmic acidification caused by anoxia in plants [73,74] while decreasing fermentative enzymes such as lactate dehydrogenase (LDH, EC 1.1.1.27), pyruvate decarboxylase (PDC, EC 4.1.1.1), and alcohol dehydrogenase (ADH, EC 1.1.1.1) [75]. Lower levels of lactate and ethanol in plant roots [10,75] and an increase in the ATP level were observed in NO_3_^−^-treated plants during waterlogging [75], which suggest that NO_3_^−^ is highly beneficial to reducing toxic metabolites while increasing the energy status of waterlogged plants. Antioxidants such as SOD, CAT, ascorbate peroxidase (APX, EC 1.11.1.11), and guaiacol peroxidase (POD, EC 1.11.1.7) remove O_2_^−^ and H_2_O_2_ [76,77]. Nitrate-fed plants show increased activities of antioxidants such as SOD and CAT, APX, and POD, thereby decreasing the level of H_2_O_2_ and O_2_^−^ [21,78], thus increasing tolerance to hypoxia and anoxia during waterlogging. Speedy recovery following re-oxygenation is equally important for plant growth, while NO_3_^−^ has been shown to be beneficial during hypoxia and subsequent re-oxygenation by inducing antioxidant systems in plants [79].

Hypoxia and anoxia in roots caused by flooding decrease chlorophyll content in the leaves of plants, thus decreasing the plant biomass and photosynthesis rate [11]. Nitrate is more beneficial in terms of biomass, net photosynthesis rate, chlorophyll, and protein content as compared to NH_4_^+^ and glycine [21,80]. Moreover, the concentration of metabolites such as sucrose, γ-aminobutyrate, succinate, and nucleoside triphosphate are reduced significantly in the absence of NO_3_^−^ during hypoxia in maize root [73]. Alanine aminotransferase (AlaAT, EC 2.6.1.2), via the reversible conversion of pyruvate and glutamate to alanine and 2-oxoglutarate, is involved in carbon and nitrogen metabolism [81]. The foliar spraying of NO_3_^−^ during waterlogging increases AlaAT and GOGAT activities along with an increase in amino acid in plants [82], suggesting that NO_3_^−^ is involved in regulating both glycolysis and the TCA cycle during O_2_ deficiency.

The nodulation of soybean plants with symbiotic bacteria is beneficial for plant growth. However, during hypoxia, non-nodulated soybean plants supplied with NO_3_^−^ have shown many beneficial effects such as more antioxidant and less oxidative damage through reduced ROS and H_2_O_2_ production as compared to nodulated soybean plants without NO_3_^−^ [79]. Similarly, nodulated soybean plants exposed to hypoxia decreased in biomass by 34%, while non-nodulated plants supplemented with NO_3_^−^ only decreased in biomass by 12% [83]. Moreover, plant NR is involved in nitrogen fixation, energy generation, maintaining cytosolic pH, and the metabolism of carbon and nitrogen in nodules during hypoxia in plant–microbe symbiosis [84,85,86]. This suggests that during hypoxia, NO_3_^−^ and NR are more beneficial than the symbiotic relationship alone.

Phytoglobins (Pgbs) play an important role during hypoxia and anoxia tolerance in plants. Numerous studies have reported that the increase in Pgbs during hypoxia and anoxia [87,88] and its expression are related to survival during O_2_-limited conditions in plants [87]. Nitrate nutrition during hypoxia has been beneficial in the overexpression of Pgbs [88]. During hypoxia and anoxia, NO_3_^−^ and NR are involved in ATP production through reductive pathways and the phytoglobin-NO respiration cycle [86,89]. This phytoglobin-NO respiration cycle helps in maintaining cellular bioenergetics by preventing the over-reduction of NAD and NADH [90]. Ethylene is responsible for the stability of the group VII ethylene response factor which could lead to the induction of several hypoxic genes [91], while NO_3_^−^ nutrition during hypoxia has led to the activation of the 4.5-fold induction of the ETR1 gene (*At1g66340.1*) responsible for ethylene production in plants [88]. The role of ethylene during hypoxia and anoxia tolerance could be due to its role in NO scavenging by inducing Phytoglobin 1 (Pgb1) mRNA levels, as ethylene does not increase NR activities during O_2_-limited conditions [92]. Interestingly, NO_3_^−^ is the substrate responsible for NO production, and it also plays a role in NO scavenging through inducing Pgb1 as well as ethylene biosynthesis, suggesting a more beneficial role of NO_3_^−^.

Redox imbalance during hypoxia and anoxia directly affects cellular metabolisms [93]. Various studies have reported that NO_3_^−^ supplementation to hypoxic and anoxic plant tissues can improve the redox state [16,94,95]. For example, NO_3_^−^ and NR maintain redox balance during hypoxia in cucumber (*Cucumis sativus* L.) [12]. Mitochondria are the most important organelles for survival, and their functionality is more crucial under hypoxia and anoxia [96]. In the absence of O_2_, NO_3_^−^ could act as a terminal electron acceptor in plants [69], while it also plays an important role in maintaining mitochondrial ultrastructure during anoxia. For example, in the absence of NO_3_^−^, cristae disappear, the matrix loses its electron density, and after a few hours, mitochondria completely degrade [69], while its presence protects the ultrastructure of mitochondria during hypoxia and anoxia [97]. In humans, NO_3_^−^ can protect against ischemia/reperfusion injury, reduce blood pressure, and improve oxidative phosphorylation efficiency (P/O ratio), indicating a decrease in proton leakage and membrane potential is distributed towards ATP synthesis in mitochondria [98].

Nitrate reduction via NR can delay cell death during hypoxia and delay the anoxic symptoms in plants [99], while its inhibition can significantly disturb the growth [95]. Tobacco (*Nicotiana tabacum*) mutant plants lacking NR reductase are more sensitive to O_2_ deprivation as compared to wild types by showing symptoms of rapid wilting, more ethanol and lactate production, and less ATP generation [94], suggesting the role of NO_3_^−^ is due to its reduction to NO_2_^−^. NR plays a role in the maintenance of energy status for nitrogen fixation under O_2_-limited conditions in *Medicago truncatula* nodules [100]. The use of NR inhibitors in the root system of nodulated alfalfa (*Medicago sativa* L.) results in a significant decrease in the ATP/ADP ratio under flooding and salinity stresses [5]. Waterlogging significantly degrades membrane lipids [101], while NO_3_^−^ and NR activity can delay the anoxia-induced degradation of membrane lipids in plant cells [102]. Higher expression of NR in cucumber (*Cucumis sativus*) than tomato (*Lycopersicon esculentum*) was associated with a high tolerance of hypoxia in the roots [103]. During hypoxia and anoxia, NR plays an important role in plant biology by regulating NO production by supplying electrons to NOFNiR and truncated hemoglobin [104]. The regulation of NO is critical, as it is a signaling and also toxic molecule if it is accumulated in a higher amount in a cell [105]. Overall, both NO_3_^−^and NR are involved in hypoxia and anoxia tolerance with numerous benefits, which suggests that NO_2_^−^ is also involved in the mechanisms. However, long-term O_2_ limitation would affect the NR acclivity, thus, again, questioning plants’ survival during O_2_-limitation conditions. For example, the NR level increases during O_2_ limitation conditions, while NR-mRNA remains constant during the early hours of O_2_ limitation and decreases after 48 h [99], suggesting long-term O_2_ limitation affects its activity. Moreover, NO, which is produced by NR itself, also decreases the level of NR protein through posttranslational modifications and ubiquitylation by affecting amino acids involved in binding the essential flavin adenine dinucleotide (FAD) and molybdenum cofactors [35,106]. Therefore, O_2_ limitation and a higher level of NO formation would affect NR activity after long-term hypoxia and anoxia, thus, again, affecting plants’ survival. Moreover, a higher concentration of NO_3_^−^ is reported to affect plant growth through the increased production of NO, thus increasing lipid peroxidation and the H_2_O_2_ level [107]. This dose-dependent effect of NO_3_^−^ might be due to the fact that beyond a certain level of its concentration, there would be more NO production through the reductive pathways, which could not be scavenged effectively, thus promoting ONOO^−^ formation, causing harmful effects.

## 4. Role of Nitrite during Hypoxia and Anoxia Tolerance

A well-known pathway of NO_2_^−^ metabolism in plants is its assimilation to amino acids through reduction to NH_4_^+^. However, during O_2_ deprivation, the assimilatory pathway is inhibited, and NO_2_^−^ is either accumulated in the cytoplasm [49] or reduced to NO by the NR in the cytoplasm or transported to mitochondria for reduction [108]. This is further supported by the fact that NiR is inhibited during O_2_-limited conditions [49]. Although NO_2_^−^ assimilation to amino acids is significantly reduced during hypoxia and anoxia, NR is activated, and the NO_2_^−^ level increases [16,52]. Studies suggest that NO_2_^−^ reduction to NO through reductive pathways is beneficial during hypoxia and anoxia [108].

Similar to NO_3_^−^, NO_2_^−^ can promote seed germination in plants [55,57,109]. Moreover, thermo-dependency during seed germination was lowered in the presence of NO_2_^−^ [61]. During low O_2_ levels in mitochondria, NO_2_^−^ can regulate the surrounding O_2_ concentration through the production of NO [53]. Exogenous NO_2_^−^ can also reduce both ethanol and lactate production [110] and can minimize the acidification of cytoplasm in plants during hypoxia and anoxia [74]. Similarly, the role of NO_2_^−^ in the protection of mitochondrial structures and functions has been well documented. NO_2_^−^ supplementation during O_2_-limited conditions to the mitochondria isolated from roots of pea (*Pisum sativum*) shows better mitochondrial integrity, the energization of the inner mitochondrial membrane, increased ATP synthesis, and decreased production of ROS and also decreased lipid peroxidation [111]. Hypoxia and anoxia can degrade the activities of complex I [112], while NO_2_^−^ supplementation can result in its higher levels and activities [111]. The role of NO_2_^−^ in hypoxia tolerance in humans and animals has been well documented [113,114]. It could be through its reduction to NO, as hypoxia and anoxia trigger NO_2_^−^ reduction to NO. However, a higher concentration of NO_2_^−^ can lead to membrane damage, lipid peroxidation, protein oxidation, mutation, DNA damage, and cell death [115], which could be through a higher level of NO production. So, for its beneficial role, its concentration should be regulated.

## 5. Role of Nitric Oxide during Hypoxia and Anoxia Tolerance

The role of NO in plant physiology has been described by numerous researchers. The reductive pathway of NO formation in plants is reported to be beneficial in plants as it promotes seed germination, increases biomass and root formation, increases energy status during O_2_ limitation, promotes tolerance to various biotic and abiotic stresses, and promotes the induction of different defense-related genes, and many others, as tabulated in Table 1.

Similar to NO_3_^−^ and NO_2_^−^, NO also stimulates germination in various plants species in a dose-dependent manner, i.e., low to medium NO has a positive effect, while a higher concentration inhibits germination [138,139,140,141]. The mechanism involved in seed germination by NO could be due to its capacity to reduce respiration rates and ROS levels while increasing carbohydrate metabolism and the level of amino acids and organic acids in germinating seeds [55]. The α-amylase (EC 3.2.1.1) activities of rice seed germination in the flooded condition are directly linked to seedling survival [142], while NO and GA can induce the activity of α-amylase [143]. However, the increase in activities of α-amylase by NO is time-dependent, such that at an early hour, it increases the activities, while prolonged NO exposure strongly reduces the activities [55]. This time-dependent role of NO could be due to the fact that prolonged exposure to NO could accumulate RNS which inhibit its activity. NO is involved in controlling seed dormancy through inducing the degradation of the ABI5 protein, thus enhancing ABA catabolism [144] while also increasing antioxidant enzymes [132,141]. However, a high level of NO can be toxic to cells, as it can inhibit mitochondrial respiration irreversibly [105] as well as inhibit antioxidants enzymes [145], which could explain the mechanisms of inhibiting germination by a higher level of NO.

NO production in plant cells during hypoxia enhances the survival rate [146]. During waterlogging conditions, the application of NO donor increases leaf area, plant biomass, harvest index, lint yield, and boll number in the cotton plant [136]. Similarly, the net photosynthetic rate and chlorophyll content increase, and MDA, H_2_O_2_, ADH, and PDC content decrease [136]. The role of NO in increasing the net photosynthetic rate and chlorophyll content could be due to its role in inhibiting the transcriptional activation of chlorophyll breakdown pathway genes such as SRG, NYC1, PPH, and PAO [147]. Similarly, during waterlogging conditions, NO influences both the morphological and physiological characteristics of maize seedlings such that it increases height, dry weight, and antioxidant activities while decreasing MDA content and the ion leakage ratio [148]. The reductive pathway of NO production is involved in maintaining leaf shape and size in plants by increasing the cell size, chlorophyll a/b contents, antioxidant enzymatic activity, homeostasis of ROS [149], and root elongation [131].

The mechanisms of hypoxia tolerance by NO are several. For example, NO improves H_2_S accumulation in maize seedling roots, which increases antioxidant defense, leading to the removal of excess ROS [146]. Moreover, during hypoxia and anoxia, NO is involved in ATP production through mitochondrial ETCs and the phytoglobin-NO cycle [16], thus increasing energy status. During hypoxia, NO production and the fine regulation of ROS and NO can slow down the respiration rate while preventing tissues from anoxia [53]. Nitric oxide could induce the expression of alternative oxidase (AOX) during various stress conditions [150], while its expression is associated with less superoxide generation and lipid peroxidation during O_2_ limitation conditions, while AOX also prevents nitro-oxidative stress during reoxygenation [151].

During plant–microbial symbiosis, the enzyme of nitrogen fixation, i.e., nitrogenase, is only stable and functional in O_2-_limited conditions [152]. In such symbiotic interaction, plant NR and mitochondrial ETCs are involved in NO production, while excess and low NO inhibit the nodule establishment [85], suggesting that NO should be regulated in the symbiotic relationship between plants and microbes. The nitrate-NO respiration process in root nodules of *Medicago truncatula* plays a role in the maintenance of the energy status required for nitrogen fixation [100].

Calcium ion reduces the level of ROS and increases the antioxidant enzymes in mitochondria during hypoxia by improving metabolism and ion transport in plants, thereby increasing hypoxia tolerance [153]. Similarly, exogenous calcium application can increase the biomass, net photosynthesis, stomatal conductance, and efficiency of photosystem II during hypoxia stress in plants [154]. NO can regulate Ca^2+^ in plant cells [155]. For example, plant cells treated with NO donors are reported to have a fast increase in cytosolic Ca^2+^ concentration, which was strongly reduced when treated with NO scavengers [156]. The mechanism involved in this regulation of Ca^2+^ could be that NO can increase the free cytosolic Ca^2+^ concentration by activating plasma membrane Ca^2+^ channels and inducing plasma membrane depolarization [156]. However, a higher concentration of NO is also reported to inhibit the cytosolic Ca^2+^ in human cells [157], suggesting NO should be regulated for its beneficial role.

If a plant is exposed to O_2_ deficiency for a prolonged period, the ultimate fate of the plant will be death. So, the mechanisms that could improve the O_2_ status of waterlogged plants would only benefit the plant to survive, while NO formation is also highly beneficial for improving O_2_ status through various mechanisms. For example, adventitious root formation increases plant resistance to waterlogging by increasing the inward diffusion of O_2_ [20] or even participating in photosynthesis, thus improving O_2_ status [158], while NO is reported to play a role on its formation during waterlogging [20]. Similarly, aerotropic roots can be originated in lateral roots that emerge above the water surface if waterlogging lasts for a prolonged period [159], while NO generation through reductive pathways is involved in lateral root and seminal root elongation in plants [133] that could facilitate the O_2_ supply together with aerotropic roots. Aerenchyma formation allows O_2_ diffusion from aerated to waterlogged parts of plants, while NO also plays a role in aerenchyma formation in plants [27], thus improving the O_2_ status. Nitric oxide as well as ethylene are involved in programmed cell death and aerenchyma formation during O_2_-limitation conditions [27,160]. Nitric oxide formed through reductive pathways induces the expression of aminocyclopropane-1-carboxylic acid (ACC) synthase (ACS) and ACC oxidase (ACO) genes responsible for ethylene synthesis [27]. Recently, it has been reported that auxin is involved in ethylene-dependent aerenchyma formation such that the use of an auxin transport inhibitor abolished the arenchyma formation [160]. Meanwhile, during O_2_ limitation conditions, NO donors could induce the upregulation of the auxin transporter *PIN2* gene [137], suggesting the diverse roles of NO in regulating aerenchyma formation in plants.

## 6. Adverse Effects of Nitric Oxide and Role of Nitric Oxide Scavenging on Hypoxia and Anoxia Tolerance

It is clear that NO, as the end product of the NO_3_-NO_2_-NO pathway, plays numerous beneficial roles during hypoxia and anoxia tolerance in plants. However, to be beneficial, the concentration of NO plays a critical role, while hypoxia and anoxia trigger NO production, which can be lethal to cells [105]. Some of the adverse effects of NO are summarized in Table 2. Moreover, oxidative stress caused by O_2_ limitation and the overproduction of NO during various stresses could damage major components of mitochondria [112,161] and inhibit antioxidants systems, thus accumulating ROS and RNS [22]. RNS, if accumulated more, could exacerbate more damage than ROS by triggering free radical peroxidation [162]. Increased RNS and ROS production could lead to retrograde signaling to the nucleus to regulate gene expressions [163]. Nitric oxide, through the formation of RNS, could lead to mutation, DNA damage, and cell death [161,164]. So, for the longer survival of a cell during hypoxia and anoxia, the NO produced RNS should be scavenged efficiently.

It is clear that NO scavengers work differently in plants. For example, the use of NO scavengers during low NO production have negative effects on plant growth [133], while during high NO production, the same NO scavengers have positive effects [166]. A similar role of NO has been reported in mammals [175]. Therefore, the optimum level of NO could be different during normal and stress conditions. As a higher amount of NO is formed through the reductive pathways during the O_2_ limitation condition, it would be beneficial that some amount of NO is scavenged from cells. For example, the scavenging of NO using NO scavengers during hypoxia preserves the function of mammals’ mitochondria [176]. There may be several pathways of NO scavenging mechanisms during O_2_-limited conditions, such as NO reduction to N_2_O [24,177] and the phytoglobin-NO cycle in plants [89].

### 6.1. Nitric Oxide Reduction to Nitrous Oxide

We found very little information available on the role of N_2_O in plants (Table 3), which could be due to the fact that N_2_O is less reactive to a biological system and is readily emitted to the atmosphere. NO formation in plants is always suspected to be underestimated [16], which could be due to that fact that NO is not simultaneously measured with N_2_O. Nitric oxide can inhibit the activity of CcO either reversibly or irreversibly, such that a lower level of NO can reversibly inhibit respiration, while a higher level of NO irreversibly inhibits it due to RNS formation [26]. This reversible and irreversible inhibition of CcO could be linked to NO reduction to N_2_O, as N_2_O is also involved in the reversible and partial inhibition of respiration at the site of CcO [178,179]. Moreover, CcO is known to reduce NO to N_2_O when both NO and O_2_ levels are low, while a higher level of NO can inhibit the NO reduction process [180]. N_2_O is a relatively inert gaseous molecule, and its formation and release to the atmosphere could significantly reduce the accumulation of RNS, while the inhibition of NO reduction to N_2_O could increase the level of RNS that could irreversibly inhibit CcO. Therefore, the dose-dependent effects of NO donors could be linked to NO reduction to N_2_O, as the optimum NO level could favor N_2_O formation [180], while at a higher dose of NO donor, NO could be high, thus favoring peroxynitirte formation, and thus exerting negative effects. The use of tungsten as an NR inhibitor was reported to inhibit N_2_O formation in plants [181], while NR inhibition challenged the plants’ survival, as described in the above section, which further supports the concept that N_2_O formation also could play a role in plants’ survival strategies. Moreover, recent results suggest that NR plays critical role in NO-mediated N_2_O formation in microalgae *Chlamydomonas reinhardtii* [182]. Both NO [183] and N_2_O [184] can increase the activities of phenylalanine ammonialyase, cinnamate-4-hydroxylase, and 4-coumaroyl-CoA ligase during pathogen attack in plants while increasing total phenolic, flavonoid, and lignin content. Similarly, both NO and N_2_O are reported to slow down fruit ripening by lowering ethylene synthesis during post-harvest storage [185,186], while the role of NO depends on the optimum dose [186], suggesting that NO could be reduced to N_2_O at the optimum dose, as discussed earlier. Therefore, the similar roles of both NO and N_2_O observed in plants could be due to NO reduction to N_2_O, which need further research as, to date, there is no research measuring both NO and N_2_O simultaneously. Interestingly, not only during O_2_ limitation [187,188] but also during UV stress, plants are reported to emit more N_2_O [189], suggesting this NO reduction to N_2_O could operate during other stresses too. Moreover, the intact chloroplast of wheat was reported to produce N_2_O when supplied with NO_2_^−^ [190], which could be due to the possible reduction of NO_2_^−^ to NO and NO to N_2_O, thus reducing the toxicity of NO and protecting the chloroplast components as in mitochondria. A field study showed a positive relationship between plant N_2_O emissions and photosynthetically active radiation [191], supporting the concept of N_2_O production in chloroplasts, too. However, to date, enzymes involved in NO reduction to N_2_O in the chloroplast are not clear, which need further research.

Microbial denitrification affects the pH by the production of OH^−^ ions [195]. Moreover, NO_3_^−^ and NO_2_^−^ supplementation in plants during hypoxia and anoxia has been reported to improve cytoplasmic acidification as well as reduce the content of ethanol, which could be toxic if accumulated in higher amounts, as reported in previous sections. The mechanism behind the reduction in ethanol and lactate content and improved cytoplasmic acidification by NO_3_^−^ and NO_2_^−^ could be due to ethanol and lactate acting as electron donors during denitrification in plants and the release of OH^−^ ions during the proposed denitrification process in plants, as shown in Equation (1) [195].

5C_2_H_5_OH + 12NO_3_^−^ → 6N_2_ + 10CO_2_ + 9H_2_O + 12OH (1)



Many field-based studies have reported a positive relationship between plant N_2_O emission and respiration rate [187,191]. This positive relation could be explained by Equation (1), as denitrification (N_2_O or N_2_ or both) with a carbon source could release CO_2_, resulting in the observed positive relationship between N_2_O emissions and respiration rate in plants. A similar observation has been reported in microbial denitrification between N_2_O and CO_2_ emissions [196]. Nitric oxide reductase (Nor) in denitrifying bacteria uses NADH as a reductase, while N_2_O is an intermediate [70]. Similarly, in the case of plant mitochondria, the addition of NADH during hypoxia can increase the NO scavenging rate [197], suggesting NO is reduced to N_2_O in a similar way to bacterial denitrification. As N_2_O is a potent greenhouse gas that contributes to global warming and ozone depletion [177,198,199,200,201], understanding its formation in plants is essential. Moreover, there exist uncertainties in estimating the global N_2_O budget [187], which could be due to the fact that sources of N_2_O are not well recognized [24,187]. For example, mainly soil-microorganisms and fungi are considered as its natural sources [198], while plants are considered as a medium to transport soil-microorganisms that produce N_2_O [202]. However, in a natural environment, plant roots may face O_2_ deficiency which favors reductive pathways of NO formation along with NO reduction to N_2_O. This is supported by the fact that plants are natural sources of both NO [203] and N_2_O [187] in field conditions too. Moreover, the excessive use of N fertilizer along with waterlogging caused by heavy rainfall events under the climate change scenario could trigger both NO and N_2_O production in plants, thus increasing their atmospheric concentration.

### 6.2. Phytoglobin-NO Cycle

As stated in the above sections, the expression of Pgbs is beneficial for plants during O_2_-limited conditions, which is due to the NO scavenging mechanism. For example, during the germination of barley seeds, the scavenging of NO through the overexpression of Pgbs resulted in a higher germination rate, protein content, and ATP/ADP ratios and a lower rate of fermentation, the S-nitrosylation of proteins and S-nitrosoglutathione (GSNO) [89,204]. These lower levels of fermentation, S-nitrosylation of protein, and GSNO level in the phytoglobin-overexpressing line could indicate that NO scavenging through Pgbs results in lower stress as it is a marker of stress. However, NO scavenged through Pgbs could be partially operated. In a low NO concentration, NO_3_^−^ formed in a similar amount to NO, while in a high NO concentration, there was no increase in NO_3_^−^ [205], suggesting that the phytoglobin-NO cycle could be a limiting factor for a higher level of NO formed during O_2_ deficiency. This is further supported by the fact that during hypoxia, about 80% of NO is scavenged by mitochondria itself [206], while Pgbs are present in the cytoplasm, chloroplast, and nucleus [207]. In this case, the recently proposed denitrification ability in plants could be another mechanism of NO scavenging [24], as laboratory as well as field-based studies have reported a significant amount of N_2_O formation in plants [187,208]. However, the proposed denitrification mechanism would also be a limiting factor during anoxia or during high NO production conditions [180]. Another topic of interest would be that the NO scavenging mechanisms would operate simultaneously or one after another so that plants would benefit more from the mechanisms during O_2_ deficiency. For example, the scavenging of NO through Pgbs can reduce S-nitrosylation in plants [89]. Phytoglobin expression in algae has been involved in the synthesis of not only NO but also N_2_O [209]. Although there is no direct evidence that Pgbs could reduce NO to N_2_O, it could play a role in the phytoglobin-NO respiration cycle. The role of Pgbs could be due to its role of oxidizing NO to NO_3_^−^ during O_2_-limitation conditions, as NO_3_^−^ is the precursor of N_2_O formation in plants [24]. Moreover, phytoglobin-overexpressing mutants could maintain a lower level of NO that can facilitate NO reduction to N_2_O during hypoxia. When NO is reduced to N_2_O, N_2_O is released to the atmosphere, suggesting a beneficial way of scavenging NO from the plant system, although N is lost from plants.

## 7. Nitric-Oxide-Mediated Post-Translational Modifications and Their Roles during Hypoxia and Anoxia

Recent research suggests that NO-mediated post-translational modifications are less reported in plants during O_2_-limitation conditions [210], which could be due to the fact that NO is reduced to N_2_O and emitted to the atmosphere. This could be supported by the fact that plants emit very high N_2_O levels during waterlogging conditions, which are even more than those in soil [202], while it has been recently suggested that the N_2_O emitted from plants even in field conditions is produced in plant cells through NO reduction [187]. However, during complete anoxic conditions, this NO reduction to N_2_O would be inhibited [180]. Moreover, the NO-scavenging capacity of Pgbs would not operate during complete anoxia [211], while they contribute to produce NO [212], thus, again, increasing the level of NO, and thus promoting nitro-oxidative stress and inducing NO-mediated post-translational modification in plants (Figure 2). Some of the NO-mediated post-translational modifications and their roles in plants during O_2_-limited conditions are discussed in the section below.

### 7.1. Protein Tyrosine Nitration (PTN)

Tyrosine nitration is the addition of a nitro group at the orthro position of the phenolic hydroxyl group of tyrosine producing 3-nitrotrysoine [213,214]. OONO^−^ can react with CO_2_ to give peroxynitous acid or ONOOCO_2_^−^ which decomposes to carbonate radical and nitrogen dioxide (NO_2_) [215], which could be a major pathway of tyrosine nitration during O_2_ deficiency, rather than being NO_2_- and O_2_-mediated. Several plant proteins are a target of tyrosine nitration, which mostly decreases their activities [215,216]. In root nodules, higher NO production was correlated with the PTN of glutamine synthetase, thus decreasing its activity [217]. As PTN is a marker of nitrosative stress, we could conclude that effective NO scavenging through Pgbs and N_2_O formation pathways would reduce the PTN, thus enhancing the plants’ survival. For example, a plant inoculated with a bacterial strain that could detoxify the NO (flavohemoglobin) was reported to have the reduced PTN of plant protein [217].

### 7.2. S-Nitrosylation

S-nitrosylation needs a preliminary reaction of NO with O_2_ via the formation of higher nitrogen oxides such as N_2_O_3_ [213]. However, when the NO concentration is more than O_2_^−^, the reaction between these two favors N_2_O_3_ formation, thus leading to S-nitrosylation [218]. Hypoxia-related NO production is reported for protein’s S-nitrosylation [219]. Several plant proteins are targets of S-nitrosylation and are inhibited [213,220,221], while some could lead to an increase in resistance to oxidative stress [221]. Evidence suggests that Pgbs, catalase, ascorbate peroxidase, and CcO are the targets of *S*-nitrosylation [222]. During hypoxia, NO is involved in S-nitrosylation, targeting GSNO reductase for selective autophagy [223]. NO can be converted to GSNO by GSH, while GSNOR converts GSNO to glutathione disulfide (GSSG) and ammonia (NH_3_). Moreover, the induction of GSNOR enzyme under anoxic conditions, more expressed in Pgb knockdown plants [89], can be simultaneously linked to these NO-scavenging mechanisms and would depend on the NO concentration, as the level of nitrosylation is only partially controlled by Pgb and GSNO reductase [89]. NO can deplete GSH content in a dose-dependent manner in biological systems [162], thus reducing the antioxidants’ availability. Moreover, peroxiredoxins (Prx), which play an important role in combating ROS and ONOO^−^ reductase activity, were inhibited through S-nitrosylation in plants exposed to high NO which was produced during different stress conditions [31]. As GSNO is a physiological NO donor, it also triggers nitrosative stress in higher concentrations [224], thus increasing damage in the plant system.

### 7.3. Metal Nitrosylation

Nitric oxide forms metal-containing proteins by binding with the metal centers of metalloprotein, known as metal nitrosylation [225]. The formation of the metal–nitrosyl complex through the metal nitrosylation process can induce conformational changes in the target proteins, impacting their activity [226]. Metal nitrosylation can prevent ROS production by blocking the peroxidation of metals [221]. However, in the plant system, hemoglobin, CcO, catalase, ascorbate peroxidase, and cytosolic and mitochondrial aconitases are reported to be inhibited through metal nitrosylation [227], thus, again, affecting plants. Therefore, an optimum level of NO is critical in reducing the negative effects.

## 8. Conclusions and Future Perspectives

All of this evidence suggests that the reductive pathways of NO formation are highly beneficial, while these pathways are triggered during O_2_ limitation, which could also lead to NO toxicity. Therefore, effective NO scavenging mechanisms could help plants survive for a longer duration during O_2_ deficiency. However, considering NO scavenging mechanisms during the O_2_-limitation conditions are defense mechanisms, the higher the stress of O_2_ limitation, the more nitro-oxidative stress there is in plants. So, we could conclude that if this reductive pathway of NO formation and scavenging is finely tuned, plants could be benefited in numerous ways. However, major scavenging systems could be limiting factors which could explain why more NO production through the reductive pathways leads to plant death in the model hypothesized in Figure 2. Although there has been much advancement in understanding the reductive pathways of NO formation, we suggest that scavenging mechanisms such as NO reduction to N_2_O in plants and their role in reducing NO toxicity is not clear. The denitrification process, which is a major pathway of nitrogen cycling in an ecosystem, is considered to be present only in micro-organisms and some fungi, while we suggest that in plants, denitrification is involved in reducing NO toxicity, thus enhancing plants’ survival in hypoxic and anoxic environments. Therefore, future research could focus on the reductive pathways of NO along with N_2_O formation while exploring the role of N_2_O formation in reducing nitro-oxidative stress. Moreover, possible sites of N_2_O formation in plant cells need further investigation. Evidence suggests that not only during O_2_ limitation conditions but also during other stresses, plants could emit more N_2_O through the reductive pathways of NO formation, which needs further research. As N_2_O is a potent greenhouse gas, understanding its formation in plants would also help in addressing the current uncertainties in its global budget and implementing mitigation strategies for its global warming effects.

## Figures and Tables

**Figure 1 ijms-23-11522-f001:**
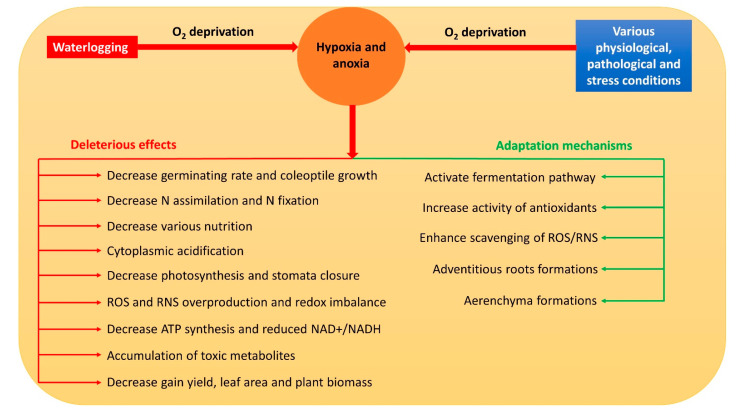
Possible causes of hypoxia and anoxia, their consequences, and defense mechanisms in response to O_2_ deficiency. Red arrows represent negative effects to plants, while green ones represent positive effects.

**Figure 2 ijms-23-11522-f002:**
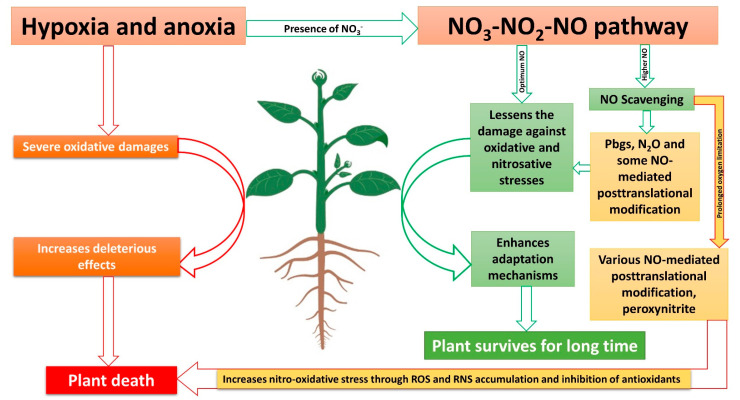
Proposed model on mechanisms of hypoxia and anoxia tolerance as well as cell death by NO_3_-NO_2_-NO pathway. The red arrows represent negative effects, while the green ones represent positive effects.

**Table 1 ijms-23-11522-t001:** Role of NO_3_-NO_2_-NO pathway during oxygen-limited conditions and other stresses on plant defense mechanisms.

	Activities/Defense Mechanisms	Conditions	References
NO_3_^−^	Maintains photosynthesis and transpiration	Waterlogging	[21]
NO_3_^−^	Protection of mitochondrial ultrastructure for a longer time	Anoxia	[97]
NO_3_^−^	Maintains membrane stability	Hypoxia	[102]
NO_3_^−^	Higher activities of antioxidant enzymes such as SOD, APX, CAT, glutathione reductase (GR, EC 1.8.1.7), and guaiacol peroxidase (GPOD, EC 1.11.1.7)	Hypoxia	[79]
NO_3_^−^	Increases the various nutrient contents	Waterlogging	[72]
NO_3_^−^	Increases seed germination rate by regulating the ABA level	Normoxia	[66]
NR inhibition	Growth is disturbed	Waterlogging	[94,95]
NO_3_^−^	Increases ATP synthesis while decreasing fermentation	Waterlogging	[75]
NO_3_^−^	Maintains the level of metabolites such as sucrose, alanine, γ-aminobutyrate, lactate, and succinate and decreases fermentation	Waterlogging	[73]
NO_3_^−^	Increases in CO_2_ assimilation, stomatal conductance, transpiration rate, and shoot biomass	Waterlogging	[78]
NO_3_^−^	UV-radiation tolerance by reducing H_2_O_2_ and malondialdehyde (MDA) and increasing plants’ height and biomass	UV stress	[116]
NO_3_^−^ and NR	Delay wilting and anoxia symptoms	Anoxia	[99]
NR-deficient mutant plant	Produces less NO that is more susceptible to bacterial and fungal attack through decreasing hypersensitive response	Pathogen attack	[117]
NO_2_^−^	Decreases fermentation that helps to reduce the toxicity of fermentative metabolites	Hypoxia	[10]
NO_3_^−^ and NO_2_^−^	Improves cytoplasmic acidification	Hypoxia and Anoxia	[73,74]
NO_2_^−^	ATP synthesis through mitochondria ETCs	Anoxia	[118]
NO_2_^−^	Protects mitochondrial structure and functions	Hypoxia	[111]
NR-dependent NO production	Defense against pathogen through rapid development of hypersensitive response and lessening the effects of clorotic lesions and bacterial infection	Pathogen attack	[40,119]
NR-dependent NO production	Involved in cold acclimation and freezing tolerance through reductions in electrolyte leakage	Cold stress	[39]
NO	Decreases the mitochondrial oxidative damages through decreased ROS content and maintained the structure and function of mitochondria through increasing mitochondrial antioxidants enzymes, improving mitochondrial Ca^2+^ homeostasis, promoting genes related to C-repeat binding factors (CBFs), while reducing the peroxidation of mitochondrial fatty acids	Cold stress	[120,121]
NO	Maintains quality, delays ripening, and enhances resistance to pathogens through increasing the activities of antioxidants, gene regulation, and suppressing ethylene production	Postharvest storage	[122]
NR-dependent NO production	Aluminum-induced ROS and lipid peroxidation are reduced, while it improves root growth during the stress through the regulation of ascorbate–glutathione cycle	Metal stress	[123,124]
NR-dependent NO production	Copper tolerance through enhanced antioxidant activities	Metal stress	[125]
NO	Improved seed germination through upregulation of α-amylase, protease, enzymes of N assimilation, and antioxidants	Metal stress	[126]
NR-dependent NO production	Salt tolerance by balancing redox and ions, reducing ROS, and increasing antioxidants	Salt stress	[127]
NO	Increases activities of antioxidants and proline content	Salt stress	[128]
NR-dependent NO production	The rapid accumulation of UV-absorbing substances such as flavonoids	UV stress	[41,129]
NR-dependent NO production	Higher photosynthetic rates and stomatal conductance, and less ROS accumulation due to higher activities of various antioxidants	Drought stress	[43]
NO	Improved photosynthesis activities and promotes growth	Drought stress	[130]
NR-dependent NO formation and induction of non-symbiotic hemoglobin	Root elongation through the activities of actin cytoskeleton and hormonal signaling	Normoxia	[131]
NR-dependent NO production	Releases tuber dormancy and sprouting via the expression of genes involved in ABA catabolism	Normoxia	[132]
NO_3_^−^ dependent NO production	Regulation of lateral root and seminal root growth by regulating auxin transport, while lateral root formation increases N uptake capacity during partial N availability	Normoxia	[133,134]
NO_2_^−^ dependent NO production	Regulates O_2_ concentration and postpone anoxia	Hypoxia	[53]
NO_2_^−^ and NO	Accelerates germination through decreasing lipid peroxidation and DNA fragmentation in germinating seeds	Physiological hypoxia	[55]
NO	Decreases cell membrane injuries and increases stomatal conductance and transpiration rate as compared to the control	Waterlogging	[135]
NO_3_^−^, NO_2_^−^ and NO	Breaks dormancy in seeds through NO signaling	Normoxia	[109]
NO	Increases biomass and lint yield of cotton plants through reduced lipid peroxidation, the expression of waterlogging tolerance-related genes, and increasing photosynthesis process	Waterlogging	[136]
NO	Enhances adventitious root formation	Waterlogging	[20]
NO	Regulates genes belonging to phytohormones, Cytochrome P450 encoding genes (*CYP72A14* and *CYP707A3*) that regulate ROS and genes related to cell wall synthesis, modification, and degradation	Hypoxia	[137]

**Table 2 ijms-23-11522-t002:** Adverse effects of a higher level of NO in plants. The high level of NO was achieved through a higher dose of NO donor or using NO-overproducing mutants or hypoxia plus NO donors.

Effects of Higher Level of NO	References
Decreases the root growth through DNA damage, induces cell cycle arrest and inhibits primary root growth by affecting root apical meristem activity and cell elongation.	[165,166]
Delayed flowering, retarded root development, and reduced starch granule formation through S-nitrosylation modification.	[167]
Cell death through increased electrolyte leakage, cell wall degradation, cytoplasmic streaming, and DNA fragmentation.	[27]
Decreases the expression of cyclins (CYC) and Cyclin-Dependent Kinases (CDKs), resulting in the downregulation of cell cycle progression.	[168]
NO can generate peroxynitrite, which is a mediator of cytochrome c loss, protein oxidation and nitration, lipid peroxidation, mitochondrial dysfunction, damage DNA, and cell death.	[26,169]
NO can inhibit antioxidants such as catalase, glutathione peroxidase (GPX), and ascorbate peroxidase in a reversible way and peroxynitrite in an irreversible way.	[145,170]
NO can change the redox state and promote cell death.	[33]
Inhibits lateral and primary root growth through reduced cell division and the expression of the auxin reporter markers *DR5pro:GUS/GFP*.	[166,171]
Inhibits growth of tobacco plants through peroxynitrite formation and tyrosine nitration.	[172]
Inhibits seed germination, while the scavenging of NO alleviates the effect.	[139]
Inhibits the shoot growth and decreases the chlorophyll contents of the plants.	[173,174]

**Table 3 ijms-23-11522-t003:** Beneficial activities of N_2_O in plants.

Beneficial Activities of N_2_O	Reference
Using post-harvest technology, the storage of fruits under N_2_O can lower ethylene production and slow the ripening of fruits.	[185,192]
N_2_O can increase resistance to pathogens by improving the accumulation of total phenolic, flavonoids, and lignin, as well as increase the activities of key enzymes in the metabolism of phenylpropanol.	[184]
Can inhibit the browning activities of enzymes such as polyphenol oxidase (PPO) and/or peroxidase (POD) and delay browning in fruits.	[193]
Can delay decay, lower the respiratory rate, and maintain the quality of fruit.	[194]

## Data Availability

Not applicable.

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
