# Peer review of "Nitrate–Nitrite–Nitric Oxide Pathway: A Mechanism of Hypoxia and Anoxia Tolerance in Plants"

_ijms, 2022, doi:10.3390/ijms231911522_

Round 1

Reviewer 1 Report

General comments:

The manuscript titled Nitrate-nitrite-nitric oxide pathway: a mechanism of hypoxia and anoxia tolerance in plants”, reviewed Oxygen (O2) biochemical processes in plants. Various biotic and abiotic factors cause O2 deprivation leading to hypoxia and anoxia in plant tissues. To survive under hypoxia and/or anoxia, plants deploy various mechanisms such as fermentation path, reactive oxygen species (ROS), reactive nitrogen species (RNS), antioxidant enzymes, aerenchyma, and adventitious root formation. In this review, the author highlights the role of reductive pathways of NO formation that lessens the deleterious effects of oxidative damages and increases the adaptation capacity of plants during hypoxia and anoxia. Meanwhile, overproduction of NO through the reductive pathways during hypoxia and anoxia leads to cellular dysfunction and cell death. And a better understanding of the molecular mechanisms involved in reducing NO toxicity would not only insight its role in plant physiology, but also address the uncertainties seen in the global N2O situation. The review was well-written. However, I have some suggestions, as shown below:

1, The manuscript is in a nice flow and easy to read, and it was well-organized with each subtitle and summarizes the recent update with recent studies. The review is a conclusion of studies, in part 8, should be outlook and future aspect.

2, All the ref. seems messed up, in the main text, the style is [xxx et al. 2020]; it is my first time to see this type, it should be (xxx et al. 2020) or [1], the line 36 [zhou et al. 2020] or [1]?  Please carefully check the ref. style required by the instruction to authors.

3, Figure1 can adjust the front and color to make it better and the legend should be as detailed as possible like the color indicates what. Figure 2 looks nice, but highly recommend modify a little bit, the plant can be bigger in the center, and Hypoxia and anoxia, NO3-NO2-NO pathway can be smaller. The arrows can be smaller, and all the figures can be aligned well

Minor revision:

Line 22: there is a therefore in line 19, please with alternative words here, same with line 28, too many therefore.

Line 103: what is NR means?  It appears in line 130, please write the full name when it first appears and then use NR instead of all.

Line 275: what is ONOO-?

Line 301: additional space before The role

Line 474: Can this be shown alone, or does it need a separate figure?

Line 504: GSNO is first 

Author Response

Thank you very much for your suggestions and corrections, we have revised the manuscript accordingly. The response to the comments are in the word file below.

Reviewer 2 Report

Authors prepared an intersting review concerning the role of NO in response to anoxia/hypoxia in plants. The study is important to this particular plant research area.  Authors used numerous articles, related to analysed field, however some novel research should be also included. Following corrections should be included before the article publication:

Following specific comments should be addressed:

Section 1.

Authores should also shortly mention which signaling pathways that are associated with the plant response to hypoxia/anoxia (for example ethylene or JA) could be affected by NO- is there any synergy or antagonism between them? More precisely, Authors could for example shortly describe metabolic pathways regulated by S-nitrosylation (Jagadis-Gupta et al., Molecular Plant, 2022; Manrique-Gil et. al., J. Exp. Bot. 2021).

Section 2.

Authors should also describe the role of alternative oxidases in regulation of NO level (Kurami et al. J. Exp. Bot. 2019 and related works) .

Authors should shortly describe the role of NO-forming nitrite reductase in  biosynthesis NO (Tejada-Jimenez et. al. 2019, Plants, Astier et. al., 2018 J. Exp. Bot., and related works).

Shortly describe routes/enzymed participating in NO degradation.

Section 3.

Authors should shortly describe how the nitrate reductase is regulated; for example at gene expression or post-translationally in context of oxygen or nitrate level. Therefore Authors could show mechanism of this regulation (Chamizo-Ampudia et al., Trends in Plant Sciences., 2017 and related works).

Section 7

Authors could add information related to nitrosylation of transition metals (Manrique-Gil et. al., J. Exp. Bot. 2021)

Other:

Authors should check if all abbreviations are explained when used for the first time.

Author Response

Thank you very much for the suggestions and corrections made for the manuscript. We have revised the manuscript according. The responses to your comments are in the word file, attached in the system.

Reviewer 3 Report

The manuscript is a review of the nitric oxide and nitrogen assimilation pathway under low oxygen conditions. The manuscript has been well organised and provides an extensive literature review.

Please avoid obvious statements such as "Plants require oxygen (O2) for survival....).

I would suggest improving the manuscript with a specific description of the enzymatic biosynthetic pathway of nitric oxide under hypoxia or anoxia. 

Moreover, the relationship between ethylene and NO should be better detailed considering the activation of Programmed Cell Death and aerenchyma formation in roots of different plant species (rice and maize). The enzymes should report the identification EC number.

Author Response

Thank you very much for the suggestions and correction made for our manuscript. We have revised it accordingly. Please see the revised version of the manuscript. the responses are also attached as a word file in the system.

Reviewer 4 Report

This review is beautifully written, I think it can be published with minimal edits. I think that it is worth explaining in more detail the facts that relate to processes in animals (for example, lines 247-249 and 301-302).

Author Response

Thank you very much for you beautiful comments for the manuscript. We revised it accordingly. Please see the revised manuscript and the responses attached in the system. 

Round 2

Reviewer 2 Report

Authors corrected all points suggested by reviewer. Manuscript is significantly improved and may be published. Small typographical errors as lack nr of  vol. or nr of pages in reference nr 32 may be corrected at later stages of manuscript preparing.